# Comparing In Vitro Virucidal Efficacy of Commercially Available Mouthwashes Against Native High-Risk Human Papillomavirus Types 16 and 18

**DOI:** 10.3390/microorganisms13040734

**Published:** 2025-03-25

**Authors:** Samina Alam, Jesus Avila, William Barrett, Craig Meyers

**Affiliations:** 1Department of Microbiology and Immunology, The Pennsylvania State University College of Medicine, Hershey, PA 17033, USA; salam@pennstatehealth.psu.edu (S.A.); jva5502@psu.edu (J.A.); 2Department of Radiation Oncology, University of Cincinnati College of Medicine, Cincinnati, OH 45219, USA; barretwl@ucmail.uc.edu

**Keywords:** Human Papillomavirus, high-risk HPV, HPV16, HPV18, mouthwash, oral rinses, head and neck squamous cell carcinoma (HNSCC), oropharyngeal squamous cell carcinoma (OPSCC), virus infectivity, oral HPV infections

## Abstract

The rising incidence of oropharyngeal cancer caused by high-risk Human Papillomavirus (HPV) type 16 and HPV18 in the U.S and other developed countries is an important public health issue. This has been attributed to changes in sexual behavior, including the practice of oral sex, which may expose individuals to increased risk of acquiring oral HPV infection. The incidence of oral HPV infections highlights the role of the oral cavity as an important anatomical site in the acquisition and transmission of high-risk HPVs. Generally, the use of mouthwash/oral rinses have focused on targeting the oral bacteriome, and could additionally be formulated for managing the oral virome. Here, we examined virucidal properties of common over-the-counter antibacterial mouthwash products against native HPV16 and HPV18 virion in vitro, and downstream modification of virus infectivity. We tested oral rinses containing essential oils/alcohol, hydrogen peroxide, and cetylpyridinium chloride. Our results demonstrated greater than 90% efficacy against HPV16 inactivation, but comparatively with less efficacy against HPV18. Overall, hydrogen peroxide containing oral rinses demonstrated the best efficacy against both high-risk types, albeit with lower efficacy against HPV18. Prophylactic virucidal oral rinses targeted towards high-risk HPVs could be beneficial in reducing incidental oral HPV load, prevalence, and persistent infections.

## 1. Introduction

Worldwide, head and neck cancers are the seventh most common tumors, with an estimated annual burden of 870,000 new cases and 440,000 deaths in 2020 [1]. Head and neck squamous cell carcinoma (HNSCC) accounts for ~4.5% of cancer diagnosis and deaths in the world [2]. The burden of HNSCC varies across the world and correlates with multiple co-factors, where cumulatively ~75% of HNSCC are caused by alcohol and tobacco use [3,4,5]. Studies have shown that oral infection with Human Papillomaviruses (HPVs) is associated with an increased risk of developing a 25% subset of head and neck cancers [6,7], including oropharyngeal squamous cell carcinomas (OPSCC) [8,9,10,11,12] and oral potentially malignant disorders [13,14,15]. Oropharyngeal squamous cell carcinomas are cancers arising from the base of the tongue subsites, walls of the oropharynx and soft palate, and tonsils [16]. Among oropharyngeal subsites, the tonsils and base of the tongue have the highest prevalence of HPV infection and cancers [7,17]. A recent report estimates that over 80% of patients with OPSCC test positively for infection with high-risk HPV genotypes [18,19], that are also the causative agents of HPV-related cervical and anogenital cancers [8,13,14].

High-risk oncogenic HPV are linked to nearly 5% of all cancer incidences, including vaginal [20], vulvar [21], cervical, and oropharyngeal cancers [22,23]. While HPV is known as the primary cause of cervical cancer, incidence of HPV related OPSCC in the U.S. has now surpassed that of cervical cancer [6,24,25,26,27] and is the most common HPV-related cancer [27]. HPV is a small, non-enveloped virus with circular, double-stranded DNA genome of approximately 8 kb [28]. More than 200 HPV types have been identified, all of which infect and complete their productive life cycle in either cutaneous or mucosal epithelia [15,29,30]. Of the high-risk HPV types 16, 18, 31, 33, 35, 39, 45, 51, 52, 56, 58, and 59 [31], HPV16 is responsible for 85–96% of HPV-positive OPSCC cases [25,32], and, to a lesser extent, HPV18 (3%) [17,32,33], while the remainder of HPV + HNSCC cases are caused by HPV33, HPV35, HPV52, HPV45, HPV39, HPV58 [6,7]. HPV16 is also the most predominant genotype detected in cervical infections, followed by HPV18, which together are the causative agents in over 70% of cervical cancer in the world [34,35,36]. About 80–90% of HPV infections are transient and clear spontaneously within 24 months after first detection [37,38]. Persistent infection with oncogenic high-risk HPV types 16 and 18, is the major cause for neoplastic transformation of oropharyngeal cancers [39,40]. HPV16 has the lowest clearance rate and is the most common genotype to persist in the oral and oropharyngeal mucosa [41,42]. Oral-genital sex is an established risk factor for acquisition of HPV infections between infected oral and genital mucus membranes [43,44,45] and development of HPV + OPSCC, with a strong association between number of lifetime oral sex partners and incidence of the disease [46,47]. Oral HPV acquisition is also positively associated with number of recent oral sex and open mouth kissing partners than with the number of vaginal sex partners [44]. The most frequently detected genotype among both men and women is the high-risk HPV16 genotype [48], with a recent cross-sectional study accounting for 12.4% of all infections among men and 8.6% of infections among women in a general U.S. population [49]. Most oral HPV infections are cleared by the immune system, but risk of long-term virus persistence in the epithelia of the upper respiratory tract could lead to development of invasive cancer within 10 years [48,50,51,52,53]. The risk of HPV spread by droplets or by aerosols is thought to be small as synchronous infection between relatives or household members has not been studied [54]. However, the relevance of saliva as a mode of oral HPV transmission [55] is supported by multiple studies [56].

The HPV DNA has been detected in exfoliated oral squames from healthy individuals [57]. A systematic review on oral HPV prevalence and asymptomatic infections in the US population included 18 studies with 4581 healthy adults showed a prevalence of 4.5% for any oral HPV type, 3.5% for high-risk HPV, and 1.3% for HPV16 [58]. Oral sex is a known risk factor, and likely primary transmission pathway for HPV acquisition in the oral cavity [44,59,60,61,62,63]. Sexual behaviors responsible for exposure to oral HPV infection are common, where 80% of the US population reported ever performing oral sex [48]. Men are more likely to have oral HPV infection than women [25,44,64]. In the U.S., representative studies estimate prevalence of oral HPV as approximately 11% in men and 4% in women [65,66]. In agreement with the higher prevalence of oral HPV in men, OPSCC is approximately three times more common in men than in women in the U.S. [6,24].

As the oral cavity is one of the anatomic sites of HPV infection, several studies have hypothesized that oral health status could be a risk factor for virus infection/persistence and oropharyngeal cancer development [67,68]. Periodontitis is a common disorder affecting greater than 40% of U.S. adults [69], and has been shown to be associated with increased risk of developing HNSCC [70]. Periodontal health can be disrupted when there are shifts in the complex interactions between the microbiota in a biofilm state and the host immune defenses [71]. As an essential component of the oral cavity, the oral microbiome contains more than 700 species of bacteria, some of which cause chronic inflammation [72,73], and thus oral dysbiosis is a risk factor for both oral HPV infection and HNSCC [70,74,75,76,77]. Chronic inflammation due to the oral microbiome could cause abnormal immune responses that interfere with the clearance of oral HPV infections, that has the potential to promote carcinogenesis [78]. Additionally, the normal oral mucosal epithelium may act as a reservoir for new HPV infections and provide a source of recurring HPV associated lesions [79]. Possible HPV reservoirs in the mouth include inflamed gingival pockets in response to oral plaque biofilms, ductal epithelium of the salivary glands, cryptal epithelium of the tonsils, border of oral cavity, and oropharynx [80]. Since the gingival pocket is the only site in the oral mucosa in which the basal cells are normally exposed to the environment, and as basal cells are known targets of HPV tropism, it has been hypothesized that this could be the site of HPV infection in the oral mucosa [80]. It is thought that dysbiosis induced local inflammation in gingival pockets and inflamed periodontal tissue may increase the probability for exposed basal cells to become infected with HPV [81], accompanied by the wound healing response, that could enhance active epithelial cell division and proliferation of infected cells, thereby aiding viral replication [61,82,83,84,85]. Although bacteria are considered the primary etiologic factors in the initiation of periodontitis regulated inflammation, host factors modulate downstream pathways of inflammatory and immune responses that control progression [86].

Antibacterial mouth rinses are widely used to improve oral hygiene, for example, by reducing dental plaque and associated inflammation of the gingiva [87]. Likewise, implementing the use of prophylactic antimicrobial rinses with virucidal activity could be important in reducing HPV load in the oral cavity to mitigate the risk of spreading the virus via contact with an infected oral mucosal membrane. Additionally, reducing the oral viral load has the potential to increase clearance of HPV infection and decrease the risk of persistence, as has been reported for anogenital HPV infections in healthy populations [88,89,90,91]. This could be important, since the lack of a clearly defined precancerous phase currently precludes the use of oral HPV testing for oropharyngeal cancer screening [92,93,94,95]. In the current study, we examined the virucidal properties of several common over the counter antibacterial mouth wash/gargling products against native HPV16 and HPV18 in vitro, in association with their ability to modify virus infectivity as an end-point measure of efficacy. We tested oral rinse products containing essential oils and alcohol, hydrogen peroxide, and cetylpyridinium chloride. Our results demonstrated that the three categories of oral rinses had greater than 90% efficacy against HPV16 inactivation, whereas only the hydrogen peroxide group of oral rinses trended towards 90% efficacy against HPV18.

## 2. Materials and Methods

### 2.1. Biosafety Measures

The experiments were carried out under Biosafety Level 2 practices and laboratory safety guidelines, approved by The Pennsylvania State University College of Medicine Biosafety Committee, protocol number BIO202300153.

### 2.2. Mouthwash and Gargling Products

The oral rinse products tested in this study were Peroxide Sore Mouth Cleanser (CVS Health, Woonsocket, RI, USA), Oragel 2X Mouth Sores Rinse (Church & Dwight Co., Ewing, NJ, USA), Listerine Ultraclean (Johnson and Johnson, Skillman, NJ, USA), Listerine Cool Mint (Johnson and Johnson, Skillman, NJ, USA), Equate Antiseptic Mouthrinse (Wal-Mart, Bentonville, AR, USA), and Crest Pro-Health (Proctor and Gamble, Cincinnati, OH, USA). As a positive control, bleach (sodium hypochlorite) was used at the manufacturer’s recommended concentration of 0.87% (8700 parts per million) (Pure Bright Germicidal Ultra Bleach, KIK International, Concord, ON, Canada). The use of this control was based on its previously demonstrated efficacy against standard laboratory stocks of native HPV16 and HPV18, in both suspension and carrier tests [96,97].

### 2.3. Cell Culture

HaCaT cells were maintained in Dulbecco’s Modified Eagle’s Medium DMEM supplemented with 10% fetal bovine serum (FBS), 0.025 mg/mL gentamicin, and 0.11 mg/mL sodium pyruvate. Mouse fibroblasts J2 3T3 cells were cultured in Dulbecco’s Modified Eagle Medium (DMEM) (Gibco cat. no. 11965-092, Life Technologies Limited, Paisley UK) supplemented with 10% newborn calf serum (NCS) and 0.025 mg/mL gentamicin. Immortalized keratinocytes stably maintaining HPV16 episomes used in this study were as follows: Cervical cell line HCK16-8 [98], foreskin cell line HPV16 wt:3 [99], and tonsil cell line HPV16HTLK (this manuscript). Immortalized keratinocytes stably maintaining HPV18 episomes used in this study were as follows: Cervical cell line HCK18 W1 (this manuscript), foreskin cell line HPV18C [100], and tonsil cell line HPV18HTLK (this manuscript). Cell lines were cultured with Mitomycin C treated J2 3T3 feeder cells as outlined [101] and maintained in E-medium. The E-Medium was prepared as per protocol previously published [101]. These cell lines have been derived in our laboratory, and are available upon request.

Cell lines described were derived using primary human cervical keratinocytes from hysterectomy tissue, primary human foreskin keratinocytes from newborn circumcision tissue and primary tonsil keratinocytes from tonsillectomy tissue. Primary cells were isolated and maintained as previously described [102]. The use of primary human tissues to develop immortalized cell lines was approved by the Institutional Review Board at the Pennsylvania State University College of Medicine and by the Institutional Review Board. Discarded, de-identified tissues were exempt from the requirement of informed patient consent. Primary keratinocytes were maintained in 154 Medium (cat. no. M154500, Cascade Biologics, Portland, OR, USA) supplemented with Human Keratinocyte Growth Supplement Kit (cat. no. S001K, Cascade Biologics, Portland, OR, USA). Full length HPV16 and HPV18 genomes were electroporated into low passage primary keratinocyte monolayer cultures, and keratinocyte lines shown to stably maintain HPV16 and HPV18 genomes were selected for growth in organotypic cultures to produce HPV16 and 18 virus stocks. The cloned HPV16 genome is the HPV16:114B prototype European variant, characterized by Dr. M. Dürst [103,104]. The cloned HPV18 genome, which is the HPV18 prototype (variant unknown), was a kind gift from Dr. H. zur Hausen [105].

### 2.4. Growing HPV16 and HPV18 Organotypic Raft Cultures

Raft cultures were grown as previously described [98,99,106]. Briefly, mouse fibroblast J2 3T3 were trypsinized and resuspended in 10% reconstitution buffer, 10% 10X DMEM (Dulbecco’s Modified Eagle Medium, cat. no. 12100-061, Gibco, Life Technologies, Grand Island, NY, USA), 2.4 μL/mL of 10 N NaOH, and 80% rat-tail type 1 collagen (Corning, NJ, USA). Cells were added at a concentration of 6.25 × 10^5^ cells per raft. The mixture was then aliquoted into 6 well plates at 2.5 mL per well and incubated at 37 °C for 2–4 h to allow solidification of the collagen matrices. Two mL E-medium was then added to each well to allow the matrix to equilibrate. Immortalized cervical keratinocytes stably maintaining HPV16 genomes (cell line HCK16-8) and immortalized foreskin keratinocytes stably maintaining HPV18 genomes (cell line HPV18C) were cultured with J2 3T3 feeder cells and maintained in E-medium and further used for growing the standard 20-day organotypic cultures. Keratinocytes from each cell line were seeded (1 × 10^6^ cells) onto collagen matrices. Following cell attachment and growth to confluence overnight, the matrices were lifted onto stainless steel grids and fed with E-medium supplemented with 10 μM 1,2-dioctanoyl-*sn*-glycerol (C8:O; Sigma Chemical Company, St. Louis, MO, USA) via diffusion from below, as previously described [98]. Raft cultures were allowed to stratify and differentiate for 20 days. Raft cultures were fed every other day, until harvesting the epithelial tissue on day 20. Mature virus particles were harvested from tissues as previously described [99]. We have reported that 20-day raft tissues are optimal for the in vitro production of highly stable native virus particles [99]. Virus stocks were prepared, and titers were determined as previously described [98] and as outlined below.

### 2.5. HPV16 and HPV18 Virus Stocks Preparation

Raft tissues were harvested and virus stocks prepared as described [98,99,106]. For preparing virus stocks, two raft tissues were Dounce homogenized in 500 μL of phosphate buffer (0.05 M sodium phosphate [pH 8.0], 2 mM MgCl_2_). Homogenizers were rinsed with an additional 250 μL of phosphate buffer. Non-encapsidated viral genomes were digested by the addition of 1.5 μL (375 U) of benzonase to 750 μL of virus preps, followed by incubation at 37 °C for 1 h. Samples were adjusted to 1 M NaCl by adding 188 μL of ice-cold 5 M NaCl. Samples were further vortexed and centrifuged at 4 °C at 10,500 rpm for 10 min. The supernatants or virus stocks were stored at −80 °C for further analysis.

### 2.6. Titering HPV16 and HPV18 Virus Stocks

Virus titers were measured using qPCR-based DNA encapsidation assay as previously described [98,99,106]. To detect endonuclease-resistant genomes in CV stocks, the following method was used. Briefly, viral genomes were released from 20 μL Benzonase-treated CV stock by re-suspension in 200 μL HIRT DNA extraction buffer (400 mM NaCl/10 mM Tris-HCl (pH 7.4)/10 mM EDTA (pH 8.0)), 2 μL 20 mg/mL proteinase K, and 10 μL of 10% SDS, and rocking for 2–4 h at 37 °C. Following digestion, the DNA was extracted twice using phenol-chloroform-isoamyl alcohol (25:24:1), followed by extraction in an equal amount of chloroform. DNA was ethanol precipitated overnight at −20 °C. Samples were centrifuged, and the DNA pellet was washed with 70% ethanol, dried, and resuspended in 20 μL of Tris-EDTA overnight. To quantify viral genomes, a Thermo Scientific Maxima SYBR Green qPCR kit was utilized. Amplification of the HPV16 E2 open reading frame (ORF) was performed using 0.3 μM of forward primer HPV16E2-5′ and HPV16E2-3′ (Appendix A). Amplification of the HPV18 E2 open reading frame (ORF) was performed using 0.3 μM of forward primer HPV18E2-5′ and HPV18E2-3′. Amplification of the E2 ORF of serially diluted pBSHPV16 and pBSHPV18 DNA, ranging from 10^8^–10^4^ copies/μL was used to generate the standard curve. A Bio-Rad iQ5 Multicolor Real-Time qPCR machine and the CFX Manager software version 3.0 were utilized for PCR amplification and subsequent data analysis respectively.

### 2.7. Disinfection Procedure

The disinfection protocol was followed, as we have previously published in multiple studies [96,97,100,107,108]. We have also presented a flow-diagram schematic of this protocol in Appendix A. For each oral rinse product, an aliquot of 1 × 10^7^ HPV16 or HPV18 virions were mixed with freshly diluted 5%FBS solution in PBS up to a total volume of 50 μL in a 1.5 mL tube. The 5% FBS solution serves as an organic load or soil to simulate physiologic conditions in the oral cavity. To this mixture, 950 μL disinfectant was added, mixed thoroughly, and incubated for 30 s contact time for efficacy, at room temperature. Then, 1 mL of an appropriate neutralizer was added to the virus/disinfectant solutions. The neutralizer used for Peroxide Sore Mouth Cleanser (CVS Health) and Oragel 2X Mouth Sores Rinse (Church & Dwight Co.) was catalase (200 U/mL in sterile water). The neutralizer used for Listerine Ultraclean (Johnson and Johnson), Listerine Cool Mint (Johnson and Johnson), and Equate Antiseptic Mouthrinse (Wal-Mart) was the HaCaT growth medium. The neutralizer used for Crest Pro-Health was catalase, as the product listed hydrogen peroxide as an “inactive” ingredient. As a positive control, sodium hypochlorite was used at the manufacturer’s recommended concentration pf 0.87% (8700 parts per million) (Pure Bright Germicidal Ultra Bleach, KIK International). The neutralizer for sodium hypochlorite was sterile 7% glycine (wt/vol) solution. The neutralizer used for untreated HPV16 and HPV18 virus as negative controls was HaCaT medium. The solutions were then centrifuged in Amicon Ultra centrifugal filters 100,000 molecular weight cut-off (MWCO; Millipore, Tullagreen, Ireland) at 4000 rpm for 10 min. The filters were washed with 2 mL HaCaT medium and centrifuged at 4000 rpm for 10 min. These steps were repeated four times. Finally, 800 μL HaCaT medium was added to each volume remaining in the filter, and the filter was gently scraped with the pipette tip and liquid pipetted up and down to ensure good virus recovery, ending in a total volume of 1000–1200 μL. The virus containing eluents were then assayed for infectivity by infecting HaCaT cells plated in 24 well plates, as described herein. The media was removed from each well and infected with the entire volume of virus containing medium. Three replicate assays were performed for each product tested. Untreated controls were included for every set of assays performed. RNA was isolated from each well and analyzed as described below.

### 2.8. RT-qPCR Infectivity Assays in HaCaT Monolayer Cultures

All infectivity studies were performed using HaCaT keratinocytes. HaCaT cells were seeded at 50,000 cells/well in 24-well plates, and infectivity assays were performed as previously described [98]. Briefly, cells were incubated with virus samples in cell culture medium for 48 h at 37 °C in 5% CO_2,_ followed by mRNA harvesting using RNeasy kit (Qiagen, Hilden, Germany). Infections were analyzed using a RT-qPCR based assay detecting absolute quantity levels of the E1^E4 splice transcript (QuantiTect Probe RT-PCR Kit). The HPV16 E1^E4 transcript was detected using 4 μM of the forward primer HPV16E1^E4-5′ and reverse primer HPV16E1^E4-3′ and using 0.2 μM of HPV16 E1^E4 fluorogenic probe (Appendix A). The TATA-binding protein (TBP) amplicons were detected using 0.125 μM primers TBP-5′ and TBP-3′, and 0.2 μM of fluorogenic probe (Appendix A). For each sample, the E1^E4 transcript abundance was normalized to TBP using infection of standard HPV16 and HPV18 laboratory stocks as controls, arbitrary designated as 1. Complete viral inactivation is considered achieved when post-disinfection infectivity assays showed equivalent or higher C_t_ values compared to uninfected controls. A Bio-Rad iQ5 Multicolor Real-Time qPCR machine and CFX Manager software version 3.0 were utilized for PCR amplification and subsequent data analysis.

## 3. Results

In this study, we tested antiviral activities of oral rinses against native HPV16 and HPV18, with virus infectivity measurement as the end-point. All our assays presented in this study were performed using native HPV16 and HPV18 stocks prepared from 20-day raft tissues that are optimal for producing highly stable virus particles [99]. We tested three mouthwash products which list essential oils and alcohol (21.6%) as their active ingredients; Listerine Antiseptic Cool Mint, Listerine Antiseptic UltraClean, and Equate Antiseptic Mouthwash (Table 1). Essential oils include eucalyptol, menthol, methyl salicylate, and thymol. We also tested two oral rinse products which list hydrogen peroxide as the active ingredients: Peroxide Sore Mouth Cleanser and Oragel 2X Mouth Sores Rinse, with the latter also stating the presence of 0.1% menthol as an active ingredient. Lastly, we tested Crest Pro-Health, which lists cetylpyridinium chloride as the active ingredient. We noted that Crest Pro-Health also listed hydrogen peroxide as an inactive ingredient. For each mouthwash product we tested a contact time of 30 s.

### 3.1. Inactivation of Tonsil Tissue Derived HPV16 (HPV16T) and HPV18 (HPV18T) Virus Stocks

We first examined the ability of the oral rinse products to inactivate HPV16 derived from growth in tonsil tissues as an appropriate model of high-risk oral HPV type found in the oral cavity. *Inactivation of HPV16T*: In line with the Federal Drug Administration (FDA) and Environmental Protection Agency (EPA) guidelines for assessing virucidal efficacy, high-level disinfectants must achieve at least a 4 log_10_ reduction in infectivity, and must achieve complete inactivation of the virus. We have previously reported the use of 0.87% bleach (hypochlorite) as a positive control for virucidal activity against native HPV preparations [97,108] and Human Coronavirus HCoV-229e, which is a less pathogenic surrogate for SARS-CoV-2 [109]. The positive control bleach achieved a 4.63 log_10_ reduction of HPV16T infectivity, in line with our previous studies, which translates to >99.99% inactivation compared to HPV16 no treatment control (Figure 1A and Table 2A). Among the three products with essential oils/alcohol as their active ingredients, Listerine Antiseptic Cool Mint inactivated infectious HPV16, with 0.787 log_10_, which translates to a trend towards 90% virus inactivation, although not reaching at least 1 log_10_ significance. (Figure 1A and Table 2A). In the same group, Equate Antiseptic Mouthrinse demonstrated <1 log_10_ reduction, which translates to <90% virus inactivation (Figure 1A and Table 2A). In comparison, Listerine Antiseptic UltraClean showed >1 log_10_ reduction, with an overall reduction of infectious virus ranging from greater than 1 log_10_ reduction to less than 2 log_10_ reduction, which translates to greater than 90% to less than 99% virus inactivation (Figure 1A and Table 2A). In contrast, among the two products with hydrogen peroxide as their active ingredients, both Peroxide Sore Mouth Cleanser and Oragel inactivated HPV16T with an overall reduction of infectious virus ranging from greater than 1 log_10_ to less than 2 log_10_ reduction, which translates to greater than 90% to less than 99% virus inactivation (Figure 1A and Table 2A). Lastly, Crest Pro-Health containing cetylpyridinium chloride inactivated HPV16 to similar levels as seen for the hydrogen peroxide group, with an overall reduction of infectious virus ranging from greater than 1 log_10_ to less than 2 log_10_ reduction, or greater than 90% to less than 99% inactivation (Figure 1A and Table 2A). It is likely that the HPV16T inactivation observed with Crest Pro-Health is due to the presence of hydrogen peroxide in the product, listed as one of its inactive ingredients, since catalase was required to neutralize the oral rinse for preventing cytotoxicity of HaCaT cells when performing the infectivity assays.

#### Inactivation of HPV18T

We then tested the efficacy of the oral rinses against native HPV18 virion derived from tonsil tissues. The positive control bleach achieved a 4.9 log_10_ reduction, in line with our previous studies, which translates to >99.99% inactivation compared to HPV18T no treatment control (Figure 1B and Table 2B). Among the three products with essential oils/alcohol as their active ingredients, all products tested showed an overall reduction of infectious virus lower than 1 log_10_ reduction, which that translates to less than 90% inactivation (Figure 1B and Table 2B). Thus, HPV18T was shown to be resistant to inactivation with oral rinses containing essential oils/alcohol as compared with HPV16T (Figure 1A). In contrast, among the three products with hydrogen peroxide as their active ingredients, Peroxide Sore Mouth Cleanser inactivated infectious HPV18T with 0.525 log_10_ reduction, which also translates to <90% virus inactivation (Figure 1B and Table 2B). In comparison, Oragel trended towards at least 90% efficacy in HPV18T inactivation with 0.806 log_10_ reduction but did not reach at least 1 log_10_ significance (Figure 1B and Table 2B). Lastly, Crest Pro-Health inactivated HPV18T with an overall reduction of infectious virus to 1 log_10_, which translates to 90% virus inactivation (Figure 1B and Table 2B).

Our results suggest that essential oils/alcohol, and hydrogen peroxide containing oral rinses inactivate HPV16T more effectively compared with HPV18T native virions propagated in tonsil tissues (Figure 1). To rule out the possibility that use of tonsil tissue to propagate the two virus types contributed to the observed differences in efficacy, we repeated the experiments using HPV16 and HPV18 virus derived from cervix and foreskin raft tissues. We reasoned that the observed differences in efficacy between HPV16 and 18 could potentially be due to cellular differences in anatomic site of the biological tissue used for propagating the virus stocks.

### 3.2. Inactivation of HPV16 (HPV16Cx) and HPV18 (HPV18Cx) Derived from Cervix Tissues

We repeated the efficacy studies of the oral rinses against native HPV16 derived from cervix tissues. *Inactivation of HPV16Cx:* The positive control bleach achieved a 4.813 log_10_ reduction of HPV16, which translates to >99.99% inactivation compared to HPV16 no treatment control (Figure 2A and Table 3A). The observed trends in cervix derived HPV16Cx associated resistance to inactivation were similar to those observed using HPV16T virus derived from tonsil tissues (Figure 1A). Among the three products with essential oils/alcohol as their active ingredients, Listerine Antiseptic Cool Mint and Equate Antiseptic Mouthrinse demonstrated similar abilities to inactivate HPV16Cx, with an overall reduction of infectious virus ranged from greater than a 1 log_10_ reduction to less than 2 log_10_ reduction, which translates to greater than 90% to less than 99% virus inactivation (Figure 2A and Table 3A). In the same group, Listerine Antiseptic UltraClean showed 0.984 log_10_ reduction, that trended towards 90% virus inactivation but did not reach at least one log significance (Figure 2A and Table 3A). In contrast, all three products with hydrogen peroxide as their active ingredients inactivated HPV16Cx with an overall reduction of infectious virus ranging from greater than 1 log_10_ to less than 2 log_10_ reduction, which translates to greater than 90% to less than 99% virus inactivation (Figure 2A and Table 3A).

#### Inactivation of HPV18Cx

Likewise, efficacy of the oral rinse products against cervix tissue derived HPV18Cx also showed a similar trend in resistance to inactivation (Figure 2B and Table 3B), as shown with tonsil tissue derived HPV18T virus (Figure 1B and Table 2B). Of the three products with essential oils/alcohol, only Listerine Antiseptic UltraClean showed efficacy against HPV18Cx that trended towards 90% inactivation with 0.741 log_10_ reduction, but did not reach at least 1 log_10_ significance (Figure 2B). For the group of oral rinses containing hydrogen peroxide, a similar pattern of infectivity reduction was observed for HPV18Cx, as shown for HPV18T (Figure 1B and Figure 2B). Use of Peroxide Sore Mouth Cleanser and Crest Pro-Health inactivated HPV18Cx and trended towards 90% inactivation but did not reach at least 1 log_10_ significance (Figure 2B and Table 3B). In comparison, Oragel 2X Mouth Sores Rinse inactivated HPV18Cx with reduction of infectious virus ranging from greater than 1 log_10_ to less than 2 log_10_ reduction, which translates to greater than 90% to less than 99% virus inactivation (Figure 2B and Table 3B).

### 3.3. Inactivation of HPV16 (HPV16F) and HPV18 (HPV18F) Derived from Foreskin Tissues

Lastly, we tested the efficacy of the oral rinses against native HPV16 derived from foreskin tissues. *Inactivation of HPV16F:* The positive control bleach achieved a 5.22 log_10_ reduction of HPV16F infectivity, which translates to >99.99% inactivation compared to HPV16F no treatment control (Figure 3A and Table 4A). Among the essential oils/alcohol group of oral rinses, only Listerine Antiseptic UltraClean showed reduction of infectious virus ranging from greater than a 1 log_10_ reduction to less than 2 log_10_ reduction, which translates to greater than 90% to less than 99% virus inactivation (Figure 3A and Table 4A). In contrast, treatment of HPV16F with Listerine Antiseptic Cool Mint and Equate Antiseptic Mouthrinse was similar to untreated control virus (Figure 3A and Table 4A). Oral rinse products in the hydrogen peroxide group, Peroxide Sore Mouth and Oragel, inactivated HPV16F with a reduction of infectious virus ranging from greater than 1 log_10_ to less than 2 log_10_ reduction, which translates to greater than 90% to less than 99% virus inactivation (Figure 3A and Table 4A). In comparison, Crest Pro-Health inactivated HPV16T with reduced efficiency, with 0.485 log reduction, and therefore did not reach at least 1 log_10_ significance (Figure 3A and Table 4A). Overall, the observed trends of HPV16F inactivation were similar to those observed using HPV16T derived from tonsil tissues.

#### Inactivation of HPV18F

Efficacy against native foreskin tissue derived HPV18 was also tested. The positive control bleach achieved a 4.083 log_10_ reduction in line with our previous studies, which translates to >99.99% inactivation compared to HPV18F no treatment control (Figure 3B and Table 4B). All oral rinse products tested showed an overall reduction of infectious virus lower than 1 log_10_ reduction, which translates to less than 90% inactivation (Figure 3B and Table 4B). HPV18F treated with Listerine Antiseptic Cool Mint was shown to be resistant to inactivation with 0.273 log_10_ reduction in viral infectivity. This was followed by Equate Antiseptic Mouthwash, and Crest Pro-Health, as both were only marginally better at inactivating HPV18, with 0.519 log_10_ and 0.655 log_10_ reduction, respectively (Figure 3B and Table 4B). Listerine Antiseptic UltraClean, Peroxide Sore Mouth Cleanser, and Oragel 2X Sore Mouth Rinse trended towards at least 90% efficacy in HPV18 inactivation, with 0.825 log_10_, 0.858 log_10_ and 0.944 log_10_ reduction, respectively (Figure 3B and Table 4B), but did not reach at least 1 log_10_ significance. Thus, the observed trends of HPV18F inactivation using the two groups of oral rinses were similar to those observed using HPV18T derived from tonsil tissues.

Overall, our results show that essential oils/alcohol, and hydrogen peroxide containing oral rinses inactivate HPV16 more effectively compared with HPV18 native virions. Furthermore, these differences observed between the two high-risk types are not biased due to the use of tissue from different anatomic sites chosen to propagate the virus utilizing in vitro raft tissues (Figure 1, Figure 2 and Figure 3). We present in Figure 4 a summary of the disinfection fold-change against the two high-risk HPV types as averages obtained using tonsil, cervix and foreskin derived HPV16 and HPV18 respectively (Figure 1, Figure 2 and Figure 3). The general trends of cumulative disinfection between HPV16 and HP18 in Figure 4 mirror those presented as single tissue types in Figure 1, Figure 2 and Figure 3. Visualizing cumulative averages shows large variation in disinfection fold-changes among the different treatment groups (Figure 4) when compared to HPV stocks prepared from anatomic site specific derived cells lines (Figure 1, Figure 2 and Figure 3), albeit this was expected. 

Based on FDA and EPA guidelines for assessing virucidal efficacy, the mouthwash/oral rinse products did not achieve at least a 4 log_10_ reduction in infectivity as would be expected from high-level disinfectants that would indicate complete virus inactivation. However, in the current study, we did observe some inactivation of HPV16 and HPV18 under conditions described herein. The observed inactivation in vitro could translate to significant reduction in oral HPV load among vulnerable populations that practice prophylactic rinsing.

## 4. Discussion

In this manuscript, we present efficacy data testing for select commercially available anti-microbial mouthwash and oral rinse products against native high-risk HPV16 and HPV18 infectivity. Overall, we observed differential resistance of HPV18 to inactivation compared to HPV16 infectivity post-treatment. Our studies showed that the three groups of oral rinses containing essential oils/alcohol, hydrogen peroxide, and cetylpyridinium chloride/hydrogen peroxide, were able to inactivate native HPV16 virion between 90% and 99% post-treatment. In contrast, we showed that native HPV18 virion were mostly resistant to inactivation by the three groups of oral rinses as compared with HPV16 post-treatment. However, generally, the three groups trended towards at least 90% inactivation of HPV18. Overall, we observed that hydrogen peroxide containing oral rinses had the highest efficacy against HPV16 and HPV18. These results agree with our previous published studies showing efficacy of a 35% sonicated hydrogen peroxide mist system as a clinical disinfectant towards HPV16, although efficacy was also demonstrated towards HPV18 infectivity in that study [96]. In the current study, the products tested contained much lower concentrations of hydrogen peroxide in comparison: Peroxide Sore Mouth contained 150 mg/10 mL hydrogen peroxide, and Oragel 2X Mouth Sores Rinse contained 1.5% hydrogen peroxide (Table 1).

The peroxidant hydrogen peroxide acts by oxidizing vital cell components such as lipids, proteins, and nucleic acids [110]. In the current study, the ability of hydrogen peroxide to affect HPV16 inactivation could potentially be due to oxidation of susceptible amino acids on L1 and L2 capsid proteins, that disrupt tertiary structure, thereby resulting in loss of infectivity. Cysteine and methionine are two sulfur-containing amino acids normally found in proteins that play multiple roles in antioxidant defense and other cellular regulatory pathways [111]. Major functions of cysteine residues in proteins include redox sensing and regulation, and antioxidant defense [112]. Cysteine residues have also been shown to regulate disulfide bond formation that determines stability of the HPV capsid structure during virus maturation [113,114,115]. Methionine residues in proteins also function in antioxidant defense [116] and regulation of cellular function through reversible oxidation and reduction [117]. Methionine residues are quite readily oxidized to methionine sulfoxide [117] by oxidants that include hydrogen peroxide and hypochlorous acid [118]. We have previously reported on the oxidizing effect of hypochlorous acid regulation of HPV16 infectivity in vitro [107]. We have also reported on the ability of chlorine dioxide to inactivate both HPV16 and HPV18 [108]. Currently, the observed differences between the two high-risk HPVs to be differentially inactivated by these oral disinfectants is not understood. Our results suggest that the observed differences between the two high-risk types to be inactivated are not due to the choice of tissue from different anatomic sites used to grow the two virus types. In effect, the differences between HPV16 and HPV18 to be inactivated by the oral rinses are maintained between the tissue types used for virus growth in three-dimensional organotypic cultures in vitro.

In future, testing of other classes of disinfectants would also be important to determine efficacy towards HPV16 and HPV18, two major high-risk types that persist in the oral cavity. Chlorine dioxide is an active ingredient used in oral health products (reviewed in [119]). As mentioned above, we have also tested and demonstrated anti-viral activity of chlorine dioxide against both HPV16 and HPV18 [108]. Other classes of active compounds currently incorporated in mouthwashes include Chlorhexidine, Cetylpyridinium chloride, Fluorides, Stannous chloride, Zinc, as well as herbal compositions that include an array of plant products such as eucalyptus oil, curcumin, and licorice (reviewed in [120]).

Resistance of HPV to disinfection by certain chemical groups could also be due to the absence of a viral envelope. Others have reported that cetylpyridinium chloride could disrupt the coronavirus lipid membrane of the viral envelope through physicochemical interactions [121]. We have recently published that human coronavirus 229e are sensitive to inactivation by mouthwashes containing essential oils, hydrogen peroxide, and cetylpyridinium chloride in vitro using very low contact times [122]. Cetylpyridinium chloride has already been reported to have antiviral effects against influenza virus [123] and Herpes Simplex Virus Type 1 [124]. On the other hand, mouthwash products like Listerine®, containing four essential oils in the essential oils, thymol, menthol, eucalyptol, and methylsalicylate in a hydroalcoholic solution [125], are known to possess antiviral activity mainly by damaging viral membranes [126,127]. How essential oils, cetylpyridinium chloride, and hydrogen peroxide treatments differentially modulate HPV16 and HPV18 infectivity remains to be determined using future studies of papillomavirus capsid structure in response to treatment with oral healthcare products.

The current study utilized highly stable and infectious HPV virions produced in 20-day organotypic raft tissues, as an optimal model for measuring efficacy of commonly available oral rinses using virus infectivity as an end-point post-treatment. We previously showed that quasivirions, spontaneously assembled from artificially expressed viral proteins and DNA, showed a vastly different resistance profile to common disinfectants when compared to native HPV virions [97]. Another study examined the effect of cetylpyridinium chloride on HPV16 L1-L2 pseudovirus preparation encapsidating a GFP reporter plasmid that showed no significant inhibition using flow cytometry as a readout of infected cells [124], which is opposite of what we observed in the current study. It is important that efficacy claims be supported by data from native HPV virions, as there is a clear difference in efficacy profiles with quasivirions [97]. In addition, our experiments were performed according to standards meant to simulate conditions in the oral cavity (high protein soil, high viral titer, and absence of cleaning, as would be representative of poor oral health) in accordance with FDA testing requirements. However, it is always possible that biological factors beyond these may also have an impact.

Incidence of HPV + OPSCC has significantly increased in the past decade. Both men and women in vulnerable populations at higher risk for oral HPV infections may benefit from prophylactic anti-virucidal oral rinses aimed towards reducing oral HPV prevalence/infections. These groups include ***young adults***, primarily white men <45 years of age, of higher socioeconomic status, who are non-smokers and non-drinkers [11,84,128], and who reported multiple oral sex partners [84,129,130]. In addition, reports now show that there is a rising trend in women and the non-white population as well [131,132]. More recently, multiple studies have observed a shift in the OPSCC disease burden in ***elderly patients*** [133,134,135]. Oral HPV DNA is more prevalent in older males aged 51–60 years [49], and males aged 60–64 years, compared to younger age groups [136], which could be alluded to differences in both age-related immune compromise and infection transmissibility through sexual behaviors [136,137,138,139,140]. Risk differences for oral acquisition of HPV could also exist based on the ***sex of one’s partner*** [141]. Women engaging in same sex-behavior were shown to have approximately nine times the prevalence of high-risk HPV infections in the oral cavity compared with heterosexual women (6.2% versus 0.7%) [49]. In addition, gay, bisexual, and other men who have sex with men (MSM) have a high risk of developing oral HPV infection [44], with bisexual men showing the highest oral high-risk as well as low-risk HPV prevalence [49]. Furthermore, prevalence of oral infection with high-risk HPV16 and HPV18 was higher for MSM compared with heterosexual men [142]. These data suggest that sexual minority populations may be at increased risk for oral HPV infections and developing OPSCC, although there is a lack of evidence as to how altered oral HPV prevalence corelates with sexual orientation. Lastly, transmission of HPV and oral HPV genotype concordance among heterosexual partners showed HPV DNA concordance in 87.5% couples [143], where oral HPV concordance is defined as the simultaneous occurrence of the same type of HPV in different anatomical sites or sharing infection in sexual partners [144]. A significant population comprises the ***HIV-positive population*** that is 2.1 times more likely to harbor HPV in the oral cavity compared to HIV-negative subjects [137]. People with HIV showed oral prevalence of HPV DNA ranging between 20% and 45%, with HPV16 found in the range of 12% and 26% of those tested [145,146]. HIV-positive people also have a three-fold higher incidence of HNSCC compared to the general population, where 50% of the OPSCCs are HPV related [147]. Higher oral HPV prevalence in the HIV-positive population may be due to an increased persistence of HPV infection due to compromised immunity, or to a high incidence of oral HPV infections as a consequence of sexual behavior [148], in addition to long-term exposure to anti-retroviral therapy [149]. Lastly, recent studies identified poor oral health as an etiological risk factor for oral and OPSCC [67,68,78], both independently and synergistically with ***tobacco and alcohol*** use [150], although this link is currently inconclusive.

Studies suggest that the oral virome may be as significant in disease pathogenesis as the oral bacteriome [151,152], including periodontal disease [153]. The HPV16 viral load may correlate to an increased number of bacteria in the oral cavity [154]. Periodontal pockets are the source of inflammatory cytokines, microbes including bacteria, as well as viruses, in the saliva may provide the perfect niche for HPV infection and persistence, in addition to serving as an oral reservoir for the virus [155]. The potential role of oral hygiene was examined in a large cross-sectional study, that showed poor oral health associated with increased likelihood of any oral HPV that could modulate oral HPV persistence [156]. It is of note that HPV16 viral load in the oral cavity is associated with specific oral microbiome composition, which correlates with increased HPV copy number [154,157,158].

The high incidence of oral HPV infections tied to the practice of risky sexual behaviors highlights the role of the oral cavity in the transmission of high-risk HPVs. Clearance rates of oral HPV infection is similar to that of anogenital HPV infections in healthy populations [88,159]. Whether in cervical or oral sites, the viral load is determined by the balance between acquisition and clearance [160,161]. Increased oral viral load is associated with reduced clearance, consistent with the cervical cancer literature [162]. Recurrent episodes of contamination are important, since without prophylactic “disinfection” steps to reduce the incident oral viral load, the duration of HPV prevalence is increased, which could lead to long-term persistence and increased probability of detecting HPV DNA at any time-point [160]. Therefore, routine oral hygiene including rinsing/gargling and oral care are important for the oral cavity to mitigate the risk of spreading the virus from prevention and transmission of HPV from one person to another. Formulating oral hygiene products with antibacterial as well as virucidal activity towards HPV may reduce the incidence of oral HPV infections [70,163]. High-risk HPV16 is the most frequent type [164] to persist in the oral cavity [41], with longer persistence among men than among women [165]. Our current study suggests that peroxide containing oral rinses could be a useful virucidal candidate for further development against high-risk HPV16 and potentially HPV18 infections.

In conclusion, it should be underscored that this is a proof-of-concept study, and that in vivo activity of such oral rinse products would need to be established for evaluating their commercial potential. Further studies would be needed to evaluate the best timing for use of the mouthwash products, taking into consideration such factors as pre-exposure or post-exposure to potential virus particles contained in biological fluids, including the amount of time necessary for prophylactic rinsing/gargling for maximal disinfection of the oral cavity.

## Figures and Tables

**Figure 1 microorganisms-13-00734-f001:**
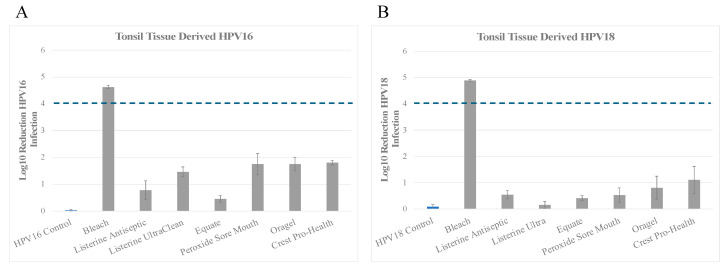
Susceptibility of HPV16 (HPV16T) and HPV18 (HPV18T) virions propagated in tonsil-derived tissues to commercially available mouthwash and oral rinses. A total of 1 × 10^7^ (**A**) HPV16 or (**B**) HPV18 particles were mixed with organic soil (5% FBS) and mixed with the oral care products indicated and disinfection protocol performed, as described in Materials and Methods. As a control for infectious virus recovery, HPV16 and HPV18 virions were mixed with soil but not treated with disinfectants. Hypochlorite was included as a positive control for disinfection efficacy. Graphs show log_10_ reduction of infectivity for each condition tested. HaCaT cells were used for the infectivity assays. The dotted line marks the FDA required 4 log_10_ reductions. FDA, Food and Drug Administration.

**Figure 2 microorganisms-13-00734-f002:**
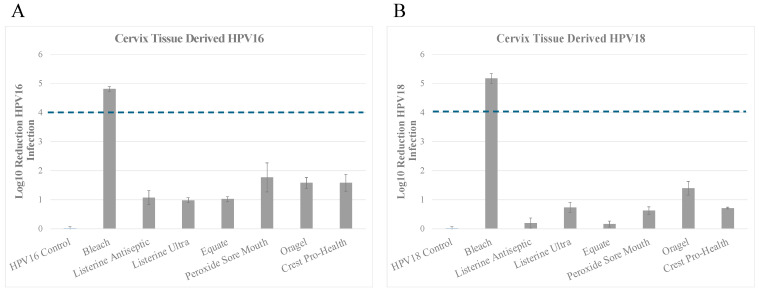
Susceptibility of HPV16 (HPV16 Cx) and HPV18 (HPV18Cx) virions propagated in cervical tissues to commercially available mouthwash and oral rinses. A total of 1 × 10^7^ (**A**) HPV16 or (**B**) HPV18 particles were mixed with organic soil (5% FBS) and mixed with the oral care products indicated and disinfection protocol performed, as described in Materials and Methods. As a control for infectious virus recovery, HPV16 and HPV18 virions were mixed with soil but not treated with disinfectants. Hypochlorite was included as a positive control for disinfection efficacy. Graphs show log_10_ reduction of infectivity for each condition tested. HaCaT cells were used for the infectivity assays. The dotted line marks the FDA required 4 log_10_ reductions. FDA, Food and Drug Administration.

**Figure 3 microorganisms-13-00734-f003:**
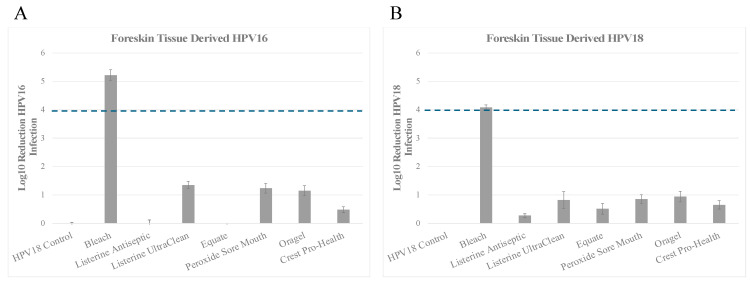
Susceptibility of HPV16 (HPV16F) and HPV18 (HPV18F) virions propagated in foreskin tissues to commercially available mouthwash and oral rinses. A total of 1 × 10^7^ (**A**) HPV16 or (**B**) HPV18 particles were mixed with organic soil (5% FBS) and mixed with the oral care products indicated and disinfection protocol performed, as described in Materials and Methods. As a control for infectious virus recovery, HPV16 and HPV18 virions were mixed with soil but not treated with disinfectants. Hypochlorite was included as a positive control for disinfection efficacy. Graphs show log_10_ reduction of infectivity for each condition tested. HaCaT cells were used for the infectivity assays. The dotted line marks the FDA required 4 log_10_ reductions. FDA, Food and Drug Administration.

**Figure 4 microorganisms-13-00734-f004:**
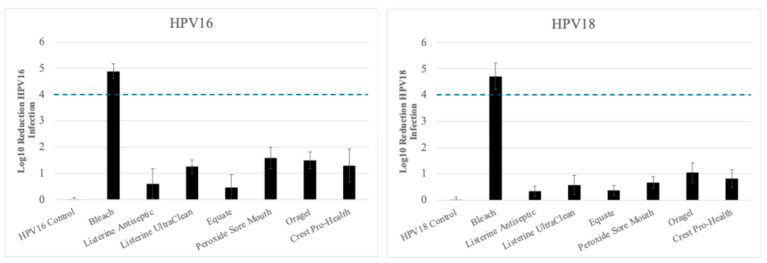
Summary trends of the disinfection fold-change against the two high-risk HPV types as averages from tonsil, cervix, and foreskin derived HPV16 and HPV18 disinfection testing, respectively (as presented in Figure 1, Figure 2 and Figure 3). The general trend of disinfection between HPV16 and HPV18 is similar to those in Figure 1, Figure 2 and Figure 3. The dotted line marks the FDA required 4 log_10_ reductions. FDA, Food and Drug Administration.

**Table 1 microorganisms-13-00734-t001:** Oral rinse products tested.

Product	Company	Active Ingredients	Inactive Ingredients
**Listerine Antiseptic** **Cool Mint**	Johnson and Johnson	Eucalyptol 0.092% Menthol 0.042% Methyl Salicylate 0.060% Thymol 0.064%	Water, Sodium Saccharin, Green 3, Alcohol (21.6%), Sodium Benzoate, Sorbitol, Flavor, Poloxamer 407, Benzoic acid
**Listerine** **Antiseptic** **UltraClean**	Johnson and Johnson	Eucalyptol 0.092% Menthol 0.042% Methyl Salicylate 0.060% Thymol 0.064%	Water, Zinc Chloride, Green 3, Alcohol (21.6%), Sodium Saccharin, Sorbitol, Sodium Benzoate, Poloxamer 407, Sucralose, Benzoic acid, Flavor
**Equate** **Antiseptic** **Mouthrinse**	Wal-Mart	Eucalyptol 0.092% Menthol 0.042% Methyl Salicylate 0.060% Thymol 0.064%	Water, Zinc Chloride, Alcohol (21.6%), Sodium Saccharin, Sorbitol, Sodium Benzoate, Poloxamer 407, Benzoic acid, FD&C Green no. 3, Flavor
**Peroxide Sore** **Mouth Cleanser**	CVS	Hydrogen peroxide 150 mg/10 mL	Water, Methyl salicylate, Sorbitol, Menthol, Propylene glycol, Sodium saccharin, Poloxamer 338, Blue 1, Polysorbate 20
**Oragel** **2X Mouth Sores Rinse Medicated**	Church and Dwight Co., Inc.	Hydrogen peroxide 1.5% Menthol 0.1%	Water, Alcohol (4.1% by vol.), Phosphoric acid, Sorbitol, Disodium EDTA, Poloxamer 338, FD&C Blue no.1, Polysorbate 20, Methyl salicylate, Sodium Saccharin
**Crest Pro-Health**	Proctor and Gamble	Cetylpyridinium chloride 0.1%	Water, Hydrogen peroxide, Glycerin, Sucralose Flavor, Poloxamer 407

**Table 2 microorganisms-13-00734-t002:** Comparing effect of mouthwash/gargles on infectivity of tonsil tissue derived HPV16T and HPV18T virus stocks.

**A.**		
Product	Tonsil Tissue Derived HPV16 (log_10_ reduction) contact time 30 s	% Inactivation
No treatment Control	0.031 ± 0.035	No change
Bleach	4.627 ± 0.062	Between >4 and <5 log_10_ (>99.99% to <99.999%)
Listerine Antiseptic Cool Mint	0.787 ± 0.348	<1 log_10_ (<90%)
Listerine Antiseptic UltraClean	1.461 ± 0.181	Between >1 and <2 log_10_ (>90% to <99%)
Equate Antiseptic Mouthrinse	0.452 ± 0.121	<1 log_10_ (<90%)
Peroxide Sore Mouth Cleanser	1.751 ± 0.398	Between >1 and <2 log_10_ (>90% to <99%)
Oragel 2X Mouth Sores Rinse Medicated	1.753 ± 0.250	Between >1 and <2 log_10_ (>90% to <99%)
Crest Pro-Health	1.810 ± 0.082	Between >1 and <2 log_10_ (>90% to <99%)
**B.**		
Product	Tonsil Tissue Derived HPV18 (log_10_ reduction) contact time 30 s	% Inactivation
No treatment Control	0.088 ± 0.083	No change
Bleach	4.89 ± 0.030	Between >4 and <5 log_10_ (>99.99% to <99.999%)
Listerine Antiseptic Cool Mint	0.545 ± 0.115	<1 log_10_ (<90%)
Listerine Antiseptic UltraClean	0.155 ± 0.141	<1 log_10_ (<90%)
Equate Antiseptic Mouthrinse	0.411 ± 0.097	<1 log_10_ (<90%)
Peroxide Sore Mouth Cleanser	0.525 ± 0.280	<1 log_10_ (<90%)
Oragel 2X Mouth Sores Rinse Medicated	0.806 ± 0.438	<1 log_10_ (<90%)
Crest Pro-Health	1.104 ± 0.516	Between >1 and <2 log_10_ (>90% to <99%)

**Table 3 microorganisms-13-00734-t003:** Comparing effect of mouthwash/gargles on infectivity of cervix tissue derived HPV16Cx and HPV18Cx virus stocks.

**A.**		
Product	Cervix Tissue Derived HPV16 (log_10_ reduction) contact time 30 s	% Inactivation
No treatment Control	0.022 ± 0.055	No change
Bleach	4.813 ± 0.086	Between >4 and <5 log_10_ (>99.99% to <99.999%)
Listerine Antiseptic Cool Mint	1.078 ± 0.241	Between >1 and <2 log_10_ (>90% to <99%)
Listerine Antiseptic UltraClean	0.984 ± 0.086	<1 log_10_ (<90%)
Equate Antiseptic Mouthrinse	1.022 ± 0.089	Between >1 and <2 log_10_ (>90% to <99%)
Peroxide Sore Mouth Cleanser	1.773 ± 0.496	Between >1 and <2 log_10_ (>90% to <99%)
Oragel 2X Mouth Sores Rinse Medicated	1.584 ± 0.183	Between >1 and <2 log_10_ (>90% to <99%)
Crest Pro-Health	1.586 ± 0.293	Between >1 and <2 log_10_ (>90% to <99%)
**B.**		
Product	Cervix Tissue Derived HPV18 (log_10_ reduction) contact time 30 s	% Inactivation
No treatment Control	0.018 ± 0.059	No change
Bleach	5.175 ± 0.171	Between >4 and <5 log_10_ (>99.99% to <99.999%)
Listerine Antiseptic Cool Mint	0.203 ± 0.177	<1 log_10_ (<90%)
Listerine Antiseptic UltraClean	0.741 ± 0.175	<1 log_10_ (<90%)
Equate Antiseptic Mouthrinse	0.167 ± 0.101	<1 log_10_ (<90%)
Peroxide Sore Mouth Cleanser	0.633 ± 0.132	<1 log_10_ (<90%)
Oragel 2X Mouth Sores Rinse Medicated	1.399 ± 0.230	Between >1 and <2 log_10_ (>90% to <99%)
Crest Pro-Health	0.716 ± 0.029	<1 log_10_ (<90%)

**Table 4 microorganisms-13-00734-t004:** Comparing effect of mouthwash/gargles on infectivity of foreskin tissue derived HPV16F and HPV18F virus stocks.

**A.**		
Product	Foreskin Tissue Derived HPV16 (log_10_ reduction) contact time 30 s	% Inactivation
No treatment Control	−0.096 ± 0.134	No change
Bleach	5.220 ± 0.188	Between >4 and <5 log_10_ (>99.99% to <99.999%)
Listerine Antiseptic Cool Mint	−0.074 ± 0.200	<1 log_10_ (<90%)
Listerine Antiseptic UltraClean	1.356 ± 0.122	Between >1 and <2 log_10_ (>90% to <99%)
Equate Antiseptic Mouthrinse	−0.099 ± 0.104	<1 log_10_ (<90%)
Peroxide Sore Mouth Cleanser	1.240 ± 0.168	Between >1 and <2 log_10_ (>90% to <99%)
Oragel 2X Mouth Sores Rinse Medicated	1.154 ± 0.179	Between >1 and <2 log_10_ (>90% to <99%)
Crest Pro-Health	0.485 ± 0.097	<1 log_10_ (<90%)
**B.**		
Product	Foreskin Tissue Derived HPV18 (log_10_ reduction) contact time 30 s	% Inactivation
No treatment Control	−0.097 ± 0.095	No change
Bleach	4.083 ± 0.090	Between >4 and <5 log_10_ (>99.99% to <99.999%)
Listerine Antiseptic Cool Mint	0.273 ± 0.073	<1 log_10_ (<90%)
Listerine Antiseptic UltraClean	0.825 ± 0.295	<1 log_10_ (<90%)
Equate Antiseptic Mouthrinse	0.519 ± 0.191	<1 log_10_ (<90%)
Peroxide Sore Mouth Cleanser	0.858 ± 0.153	<1 log_10_ (<90%)
Oragel 2X Mouth Sores Rinse Medicated	0.944 ± 0.189	<1 log_10_ (<90%)
Crest Pro-Health	0.655 ± 0.144	<1 log_10_ (<90%)

## Data Availability

Data supporting reported results can be made available upon request from the corresponding author.

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
