# Peer review of "Comparing In Vitro Virucidal Efficacy of Commercially Available Mouthwashes Against Native High-Risk Human Papillomavirus Types 16 and 18"

_microorganisms, 2025, doi:10.3390/microorganisms13040734_

Round 1

Reviewer 1 Report

Comments and Suggestions for Authors

The study provides valuable insights into the limitations and potential benefits of current mouthwash formulations in reducing HPV infectivity, while also highlighting the need for further in vivo studies to assess whether these in vitro reductions translate into meaningful reductions in oral HPV transmission or persistence.However few minor criticisms should be addressed

Minor criticicms

The manuscript speculates on differences in capsid structure (e.g., susceptibility due to cysteine or methionine residues) to explain the differential efficacy between HPV16 and HPV18. However, it does not provide direct biochemical or structural data to support these hypotheses.

• The presentation of experimental details is dense. A clearer schematic or additional summary figures could help improve readability and facilitate understanding of the protocol and the comparative results.

• While the focus on HPV16 and HPV18 is justified by their clinical relevance, evaluating additional high-risk HPV types could broaden the study’s impact

Author Response

Reviewer 1

Comments and Suggestions for Authors

The study provides valuable insights into the limitations and potential benefits of current mouthwash formulations in reducing HPV infectivity, while also highlighting the need for further in vivo studies to assess whether these in vitro reductions translate into meaningful reductions in oral HPV transmission or persistence. However few minor criticisms should be addressed

Minor criticisms

  • The manuscript speculates on differences in capsid structure (e.g., susceptibility due to cysteine or methionine residues) to explain the differential efficacy between HPV16 and HPV18. However, it does not provide direct biochemical or structural data to support these hypotheses.

Answer: Thank you for this comment. Our discussion on the observed differential efficacy between HPV16 and HPV18 is purely speculative, and we do not currently have data to support this hypothesis. Therefore, we have removed all related mentions and discussions from the manuscript.

  • The presentation of experimental details is dense. A clearer schematic or additional summary figures could help improve readability and facilitate understanding of the protocol and the comparative results.

Answer: Thank you for this comment. We have added a flow-diagram (Figure S1) for clearer understanding of the disinfection steps outlined in the Materials and Methods section. Additionally, we have now added a new Figure 4, that cumulatively summarizes the disinfection data for HPV16 and HPV18 as averages from the experiments of tonsil, cervix and foreskin derived HPV16 and HPV18 respectively (Figures 1-3). The overall observed trends of disinfection between HPV16 and HPV18 (Figure 4) mirrors those presented in Figures 1-3.

  • While the focus on HPV16 and HPV18 is justified by their clinical relevance, evaluating additional high-risk HPV types could broaden the study’s impact

Answer: Thank you for this comment. This manuscript focuses on oral health, where more than 90% of HPV infections are due to HPV16 followed by HPV18. Other high-risk HPV types in the oral cavity are rarely found to cause disease.

Reviewer 2 Report

Comments and Suggestions for Authors

Very well presented and interesting work, with potentially clinically useful results. I only have a few observations that I consider should be addressed.

Percent match: 38% (iThenticate report): The manuscript presents a high percentage of similarity with other texts, especially in some parts of the methodology. Sections should be rewritten to lower the % of similarity, since such a high % is unacceptable.

2.4. HPV16 and HPV18 Virus Stocks Preparation: Please add a reference where the production of HPV viral particles with the described technique is demonstrated.

2.6. Disinfection procedure; Add references.

2.7. RT-qPCR infectivity assays in HaCaT Monolayer Cultures: Which thermocycler did you use? Was it an absolute or relative quantity?

Show a letter of approval from a scientific or biosafety committee, or mention the biosafety measures used, in accordance with international standards. What level of biosecurity were the tests carried out under?

Although the origin is mentioned, please describe which variant HPV 16 and 18 are used. For example, the European HPV is less oncogenic or aggressive than the AA or NA variant. Just mention it.

In discussions, mention that other future studies are necessary to compare with other products that may be useful, such as chlorine dioxide (https://pmc.ncbi.nlm.nih.gov/articles/PMC7497195/).

Author Response

Reviewer 2

Comments and Suggestions for Authors

Very well presented and interesting work, with potentially clinically useful results. I only have a few observations that I consider should be addressed.

1- Percent match: 38% (iThenticate report): The manuscript presents a high percentage of similarity with other texts, especially in some parts of the methodology. Sections should be rewritten to lower the % of similarity, since such a high % is unacceptable.

Answer: Thank you for this comment. The reproducibility of our data depends on strict guidelines of our developed protocols. In turn, the methodology is written according to the protocols used. Thus, it is difficult to change the language of the methodology to lower the % similarity, as our basic protocols have not changed. We hope this explanation is helpful towards addressing your concerns.

2- HPV16 and HPV18 Virus Stocks Preparation: Please add a reference where the production of HPV viral particles with the described technique is demonstrated.

Answer: We have added the references in the Materials and Methods section.

3-   Disinfection procedure; Add references.

Answer: We have added multiple references of related disinfection studies we have published, in the Materials and Methods section.

4-  RT-qPCR infectivity assays in HaCaT Monolayer Cultures: Which thermocycler did you use?  Was it an absolute or relative quantity?

Answer: We have added the name of the thermocycler used. We used absolute values of the transcripts analyzed. We have added these information in the Materials and Methods section.

5- Show a letter of approval from a scientific or biosafety committee, or mention the biosafety measures used, in accordance with international standards. What level of biosecurity were the tests carried out under?

Answer: We have added our approved Biosafety protocol number in the Materials and Methods section. We are a BSL2 laboratory and our biosafety measures adhere to BSL2 protocols.

6- Although the origin is mentioned, please describe which variant HPV 16 and 18 are used. For example, the European HPV is less oncogenic or aggressive than the AA or NA variant. Just mention it.

Answer: We are using the prototype HPV16 and HPV18 genomes. The HPV16 genome is the European variant first characterized by Dr. Mattias Durst. The HPV18 genome was a gift from Dr. Harald zur Hausen. We do not have variant information related to the HPV18 genome. We have included related references in the Materials and Methods section.

7- In discussions, mention that other future studies are necessary to compare with other products that may be useful, such as chlorine dioxide (https://pmc.ncbi.nlm.nih.gov/articles/PMC7497195/).

Answer: We have added a paragraph in the Discussion section mentioning other active ingredients in oral rinse products, including chlorine dioxide, and their relevance to designing future studies of HPV disinfection.